# TEST-TIME GENERALIZATION FOR PHYSICS THROUGH NEURAL OPERATOR SPLITTING

## ABSTRACT

Neural operators have shown promise in learning solution maps of partial differential equations (PDEs), but they often struggle to generalize when test inputs lie outside the training distribution, such as novel initial conditions, unseen PDE coefficients or unseen physics. Prior works address this limitation with large scale multi physics pretraining followed by fine tuning, but this still requires examples from the new dynamics, falling short of true zero shot generalization. In this work, we propose a method to enhance generalization at test-time, i.e, without modifying pretrained weights. Building on DISCO, which provides a dictionary of neural operators trained across different dynamics, we introduce a neural operator splitting strategy that, at test time, searches over compositions of training operators to approximate unseen dynamics. On challenging out-of-distribution tasks including parameter extrapolation and novel combinations of physics phenomena, our approach achieves state-of-the-art zero shot generalization results, while being able to recover the underlying PDE parameters. These results underscore test-time computation as a key avenue for building flexible, compositional, and generalizable neural operators.

## 1 INTRODUCTION

Neural surrogates (de Bézenac et al., 2019; E et al., 2021; Pfaff et al., 2020; Brandstetter et al., 2022) and neural operators (Lu et al., 2021; Li et al., 2020; Kovachki et al., 2021; Raonic et al., 2023; Serrano et al., 2023) offer powerful data-driven tools for modeling spatiotemporal dynamics and systems governed by partial differential equations (PDEs). Their main limitation, however, is sensitivity to distribution shifts at test time, i.e., when the dynamics are out-of-distribution (OOD). Such shifts can arise from variations in initial conditions (Chen et al., 2024), error accumulation during autoregressive rollouts (Brandstetter et al., 2022; Lippe et al., 2023; Pedersen et al., 2025), changes in PDE parameters (Kirchmeyer et al., 2022; Koupaï et al., 2024), or fundamentally different underlying dynamics (Takamoto et al., 2022; McCabe et al., 2023; Herde et al., 2024).

We focus on the *parametric setting* (Cohen & Devore, 2015), where a neural surrogate is trained to emulate families of physical dynamics parameterized by varying coefficients, with the goal of generalizing across a range of parameter values. Our interest lies in assessing the ability of such surrogates to extrapolate—either to parameter values never encountered during training or to novel compositions of dynamics seen individually during training—while having access to only limited observation data for adaptation.

To address failures under OOD conditions, many recent frameworks (McCabe et al., 2023; Herde et al., 2024; Hao et al., 2024) adopt a *pretrain–then–finetune* paradigm. While often effective, this strategy breaks down when very limited data are available for finetuning (Koupaï et al., 2024), and the models face fundamental limitations due to their lack of compositionality, remaining constrained by the diversity of the pretraining distribution.

Meta-learning (Thrun & Pratt, 1998; Finn et al., 2017) offers an alternative, aiming to learn shared representations that can be rapidly adapted to new parameter regimes (Yin et al., 2022; Kirchmeyer et al., 2022; Koupaï et al., 2024; Nzoyem et al., 2025). However, these approaches have yet to scale reliably to diverse physical systems (Ohana et al., 2024; Morel et al., 2025), and parameter adaptation has been shown to be unstable under distribution shifts (Serrano et al., 2025).

To overcome these limitations, we propose a novel test-time adaptation approach for neural surrogates that increases expressivity and predictive quality while keeping model weights fixed. Our method approximates unobserved dynamics as a sum of neural ODE operators learned during training. The framework consists of three components: (1) a pretrained DISCO (Morel et al., 2025), a scalable framework that infers a neural ODE operator from each training trajectory and encodes it in a shared, compact latent space; (2) an efficient beam search over the discrete operators discovered during training, which identifies a suitable decomposition of the unknown dynamics; and (3) operator splitting (Strang, 1968), used both during the search and rollout to approximate the sum of differentiable terms through successive compositions. This test-time strategy comes at a higher computational cost, but enables to better adapt when faced with unseen dynamics.

We evaluate our method against existing approaches on two challenging OOD zero-shot scenarios: (1) when the PDE coefficients lie outside the training distribution, and (2) when the spatiotemporal dynamics result from combinations of physical processes that were observed only individually during training. Our results show that the proposed approach outperforms other methods in both zero-shot settings. Our key contributions are as follows:

- We propose a novel test-time generalization strategy for evolving PDEs that combines neural operators with operator splitting to approximate OOD spatiotemporal dynamics.
- We adapt a beam search procedure to efficiently combine pretrained operators, balancing accuracy and computational cost, and provide corresponding test-time scaling laws.
- We demonstrate state-of-the-art zero-shot generalization across diverse nonlinear PDEs and tasks—including parameter extrapolation and operator composition—outperforming adaptive neural operator methods and transformer-based architectures.
- Analyzing the resulting operator decompositions provide insight into the unseen dynamics by enabling zero-shot PDE parameter estimation.
- To the best of our knowledge, this is the first work to tackle test-time generalization for predicting PDEs.

## 2 RELATED WORK

**Surrogate models for PDEs.** With the goal of accelerating simulation-based workflows, and supported by growing collections of datasets (Takamoto et al., 2022; Ohana et al., 2024; Koehler et al., 2024), surrogate models (McCabe et al., 2023; Hao et al., 2024; Herde et al., 2024; Serrano et al., 2025; Morel et al., 2025) have gradually improved their pretraining performance, and achieve better results than training from scratch when fine-tuned on out-of-distribution PDEs. In contrast, our approach operates at test time without updating the model weights.

**Meta-learning strategies for dynamical systems.** Meta-learning strategies aim at rapidly adapting to new tasks (e.g., unseen PDE coefficients) by leveraging shared weights across tasks. However, many methods make restrictive assumptions, such as knowing the new PDE coefficients (Wang et al., 2022), PDE symmetries (Mialon et al., 2023), or affine predictability (Blanke & Lelarge, 2023). In GEPS, Kirchmeyer et al. (2022); Koupaï et al. (2024) adapt a shared operator to unseen physics, but this still requires fine-tuning a neural operator, which can be costly, especially in far OOD scenarios. In contrast, our approach builds on DISCO (Morel et al., 2025), which encodes the training set into a latent space of neural operators from the trajectory data only, and can generalize to unseen complex physics phenomena and PDE parameters well beyond the training range.

**Test-time strategies.** Methods to improve model performance at test time (i.e., after pretraining) have emerged with the scaling of large language models (Meta, 2025; MistralAI, 2025; OpenAI, 2025). In Best-of-N, the model generates multiple candidate outputs and selects the best one according to a predefined criterion or reward (Huang et al., 2025). Beam search extends this idea by maintaining and refining a set of promising output sequences as the model generates them (Xie et al., 2023). Similarly, our approach allocates additional compute at test time by exploring different compositions of operators seen during training and selecting the one that best fits the beginning of the test-time trajectory. To our knowledge, this is the first work to introduce test-time strategies for evolving dynamical systems governed by PDEs, and we further provide test-time compute scaling laws (Hestness et al., 2017) in this context.

## 3    PROBLEM SETTING

Data-driven models for evolving unknown PDEs are typically trained on trajectories with varying PDE class, coefficients, and initial conditions, with the goal of generalizing to unseen scenarios. While prior works have focused on novel initial conditions under a fixed PDE, we consider the more challenging setting of generalization to unseen PDE coefficients or even entirely new PDE classes. This setting relates to approaches such as MPP (McCabe et al., 2023) and DISCO (Morel et al., 2025), but we restrict the diversity of training physics to better evaluate OOD generalization.

**Parametric PDE setting.**    We consider a family of parametric PDEs of the form

$$\partial_t u = \sum_{i=1}^{K} \mu_k \, \mathcal{F}_k(u, \nabla_x u, \nabla_x^2 u, \dots),$$

where $u(x, t)$ is the solution field, $\mu = (\mu_1, \dots, \mu_K) \in \mathcal{M}$ is a parameter vector, and $\{\mathcal{F}_i\}$ denote fundamental physics operators (e.g., advection, diffusion, reaction). During training, parameters are drawn from a *sparse distribution* $P^{\text{train}}(\mu)$, where only one operator is present at a time. Concretely, each sample takes the form $\mu = (0, \dots, \mu_k, \dots, 0)$, with exactly one nonzero component $\mu_k \in \mathcal{M}_k^{\text{train}}$, restricted to a prescribed training range.

**OOD challenges.**    This setup naturally induces two distinct types of OOD scenarios at test time:

- *Parameter Extrapolation*: Parameters remain sparse but take values *outside* the convex hull of training ranges: $\mu_{\text{test}} = (0, \dots, \mu_k^{\text{test}}, \dots, 0)$ with $\mu_k^{\text{test}} \notin \text{conv}(\mathcal{M}_k^{\text{train}})$.
- *Operator Composition*: Multiple operators are simultaneously present, though each parameter still lies *within* its training range: $\mu_{\text{test}} = (\mu_1, \dots, \mu_K)$ with several $\mu_i \neq 0$ and $\mu_i \in \text{conv}(\mathcal{M}_i^{\text{train}})$.

For illustration, consider the advection–diffusion equation $\partial_t u + c \, \partial_x u = D \, \partial_{xx} u$, with advection speed $c$ and diffusion coefficient $D$. Training covers pure advection ($c \in [0, 1], D = 0$) and pure diffusion ($c = 0, D \in [0, 1]$) separately. At test time, parameter extrapolation may involve $c = 2.5, D = 0$, while operator composition may involve $c = 0.5, D = 0.3$.

**Zero-shot prediction task.**    Given this OOD setting, our task is to predict rollout trajectories in a zero-shot manner using only the observed dynamics at test time. Specifically, we observe $L$ consecutive snapshots of a test trajectory $u_{\text{test}}^{1:L}$ with temporal discretization $\Delta t$, which characterize the underlying dynamics that were never seen during training. From these observations alone, we must predict the subsequent $H$ snapshots $\hat{u}_{\text{test}}^{L+1:L+H}$ without any parameter updates or training on this specific system. Performance is evaluated using the normalized relative mean squared error (NRMSE) against the ground truth:

$$\text{NRMSE}(u_{\text{test}}, \hat{u}_{\text{test}}) = \frac{||u_{\text{test}} - \hat{u}_{\text{test}}||_2}{||u_{\text{test}}||_2} \ .$$

## 4    METHOD

### 4.1    CONSTRUCTING A DICTIONARY OF OPERATORS

The DISCO framework (Morel et al., 2025) learns to predict PDE evolution by discovering appropriate differential operators from trajectory context. It consists of two main components: a hypernetwork $\psi_\alpha$ that processes spatiotemporal context, and a small operator network $f_\theta$ that performs the actual time integration. Given a trajectory context $u^{1:L}$, DISCO operates through:

$$\hat{u}^{L+1} = u^L + \int_{L}^{L+1} f_\theta(u^t) \, dt, \quad \text{with} \quad \theta = \psi_\alpha(u^{1:L}) \, ,$$

where $\psi_\alpha$ is a transformer with learnable parameters $\alpha$, and $f_\theta$ is a U-Net whose parameters $\theta$ are dynamically generated by the hypernetwork. After pretraining, we extract a dictionary of neural operators by encoding for each trajectory $i$ from the training set: $\{f_{\theta_i} = \psi_\alpha(u_i^{1:L})\}$. To simplify notation, we denote $f_i = f_{\theta_i}$. This dictionary of operators $f_1, \dots, f_N$ will form the foundation of our test-time search strategy.

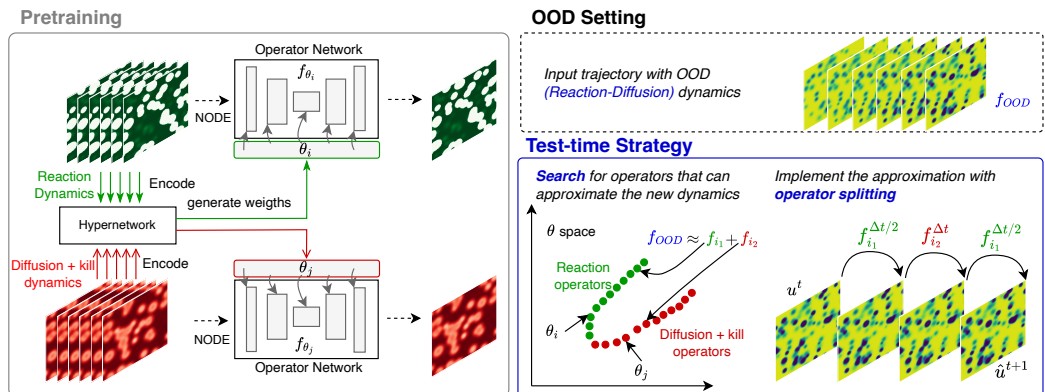

Figure 1: **Test-time generalization through neural operator splitting**. During pretraining (left), DISCO learns operators for different physics—e.g., reaction dynamics (green) and diffusion+kill dynamics (red)—with a hypernetwork generating corresponding operator weights $\theta_i, \theta_j$. At test time (right), on OOD dynamics, e.g., reaction-diffusion dynamics, our method searches the operator space $\theta$ to identify combinations approximating the new dynamics (e.g. $f_{OOD} \approx f_{i_1} + f_{i_2}$), and uses operator splitting to evolve $u^t \to u^{t+1}$ through sequential operator applications.

## 4.2 OPERATOR COMPOSITION SEARCH

Given a test trajectory $u_{\text{test}}^{1:L}$ governed by unknown dynamics, our goal is to approximate the underlying system by composing operators from our dictionary $\{f_1, \ldots, f_N\}$. We seek a subset $S = \{f_{i_1}, f_{i_2}, \ldots, f_{i_m}\}$ such that the sum $f_{i_1} + f_{i_2} + \cdots + f_{i_m}$ best approximates the test dynamics. In practice, this sum is implemented through operator splitting as detailed in Section 4.3.

**Optimization objective.** We define $\mathcal{L}(S) = \frac{1}{L-1} \sum_{t=1}^{L-1} \text{NRMSE}(u_{\text{test}}^{t+1}, \hat{u}_{\text{test}}^{t+1})$ as the prediction loss when using subset $S$, where $\hat{u}_{\text{test}}^{t+1}$ is the prediction obtained by applying operator splitting with the operators in $S$ starting from $u_{\text{test}}^t$. Our test-time adaptation seeks for the discrete minimizer of this objective $S^* = \arg\min_{S \subseteq \{f_1, \ldots, f_N\}} \mathcal{L}(S)$.

**Search Strategies.** Since an exhaustive search over all $2^N$ possible subsets is intractable, we investigate two complementary strategies that balance exploration with computational efficiency.

*Uniform Sampling*: As a baseline, we uniformly sample subsets of size $m \sim \text{Uniform}(1, M)$ by randomly selecting $m$ operators from our dictionary, where $M$ is a small maximum subset size. We evaluate $T$ random trials, giving a computational complexity of $O(T)$. See Algorithm 2 for more details.

*Beam Search*: We use beam search to efficiently explore operator combinations while maintaining computational tractability. Starting with the top-$B$ single operators ($B$ is the beam width), we iteratively expand each candidate by adding one more operator and keep only the $B$ best combinations:

$$\mathcal{B}_0 = \text{top-}B \text{ operators from } \{f_1, \ldots, f_N\},$$
$$\mathcal{B}_{m+1} = \text{top-}B \text{ from } \{S \cup \{f_j\} : S \in \mathcal{B}_m\}.$$

Here, $\mathcal{B}_0$ contains singletons, $\mathcal{B}_1$ pairs, $\mathcal{B}_2$ triples, and so on. The computational complexity is $O(BN)$ per iteration. When $B = 1$, this reduces to greedy sequential selection. To prevent excessive operator combinations, we impose both a minimum relative improvement threshold to continue the search and a maximum composition length of $M$. The pseudo-code is detailed in Algorithm 1.

## 4.3 OPERATOR SPLITTING FOR NEURAL OPERATORS

To implement the sum $f_{i_1} + f_{i_2} + \cdots + f_{i_m}$ in practice, we employ neural operator splitting. For two operators $f_1 + f_2$, Lie splitting sequentially applies each operator over the full time step: $\hat{u}^{L+1} = f_2^{\Delta t} \circ f_1^{\Delta t}(u^L)$, where $f_i^{\Delta t}$ represents integrating operator $f_i$ for time $\Delta t$. Strang splitting

---

**Algorithm 1** Beam Search Operator Composition

---

**Require:** Test trajectory $u_{\text{test}}^{1:L}$, operator dictionary $\{f_1, \ldots, f_N\}$, beam width $B$, max iterations $M$,
    Improvement threshold $\tau$
**Ensure:** Best operator subset $S^*$
  1: Initialize beam: $\mathcal{B}_0 = $ top-$B$ operators from $\{f_1, \ldots, f_N\}$ ranked by $\mathcal{L}(\{f_i\})$
  2: **for** $m = 0$ to $M - 1$ **do**
  3:    Candidates $= \emptyset$
  4:    **for** each $S \in \mathcal{B}_m$ **do**
  5:      **for** each $f_j \in \{f_1, \ldots, f_N\} \setminus S$ **do**
  6:        Add $S \cup \{f_j\}$ to Candidates
  7:      **end for**
  8:    **end for**
  9:    $\mathcal{B}_{m+1} = $ top-$B$ from Candidates ranked by $\mathcal{L}(\cdot)$
10:    **if** relative improvement $< \tau$ or $m = M - 1$ **then**
11:      **break**
12:    **end if**
13: **end for**
14: **return** $\arg\min_{S \in \mathcal{B}_m} \mathcal{L}(S)$

---

uses a symmetric pattern for higher accuracy: $\hat{u}^{L+1} = f_1^{\Delta t/2} \circ f_2^{\Delta t} \circ f_1^{\Delta t/2}(u^L)$. This reduces the approximation error from $\mathcal{O}(\Delta t^2)$ to $\mathcal{O}(\Delta t^3)$ (Strang, 1968; Holden et al., 2010). For multiple operators, we extend these patterns while maintaining computational tractability. This is to the best of our knowledge the first application of operator splitting in the context of neural PDE surrogates.

## 5 EXPERIMENTS

We evaluate our test-time search strategy on two challenging OOD scenarios, using distinct benchmarks to systematically assess the capabilities of operator composition and test-time adaptation.

We begin by describing the experimental setup and training dataset (Section 5.1). We then evaluate extrapolation performance to unseen PDE parameter ranges, demonstrating how test-time search enables robust generalization beyond the training distribution (Section 5.2). Next, we assess our method's ability to handle novel compositions of physical processes in (Section 5.3). We also analyze how our approach benefits from increased computational budget during test-time search, showing consistent performance improvements, and demonstrate its capacity for parameter identification in previously unseen dynamical systems.

### 5.1 EXPERIMENTAL SETTING

**Datasets.** We design three benchmark datasets that systematically evaluate compositional generalization capabilities across different physics regimes and spatial dimensions. Each training dataset enforces strict separation of physical processes as described in Section 3, with operators learned exclusively from trajectories containing individual physics components, never their combinations.

**1D Advection-Diffusion.** Our first benchmark focuses on linear transport phenomena governed by $\frac{\partial u}{\partial t} = D\frac{\partial^2 u}{\partial x^2} - c\frac{\partial u}{\partial x}$ on a periodic domain of length $l = 16$ with 256 spatial discretization points. Training data consists exclusively of single-physics trajectories: pure advection with speeds $c \in [0.01, 1.0]$ and zero diffusion ($D = 0$), or pure diffusion with coefficients $D \in [0.001, 1.0]$ and zero advection ($c = 0$). Each trajectory contains 100 temporal snapshots spanning $T = 10$ seconds.

**1D Combined Equation.** The second dataset examines the nonlinear advection-diffusion-dispersion equation: $\frac{\partial u}{\partial t} + \alpha\frac{\partial u^2}{\partial x} - \beta\frac{\partial^2 u}{\partial x^2} + \gamma\frac{\partial^3 u}{\partial x^3} = 0$, where $\alpha$, $\beta$, and $\gamma$ quantify the strength of nonlinear advection, diffusion, and dispersion effects, respectively. Training isolates each physical mechanism with parameter combinations $(\alpha, 0, 0)$, $(0, \beta, 0)$, and $(0, 0, \gamma)$, where coefficients are sampled uniformly from $\alpha \in [0, 1]$, $\beta \in [0, 0.4]$, and $\gamma \in [0, 1]$. We generate 8,192 training trajectories for each physics across 128 parameter configurations, with each trajectory containing

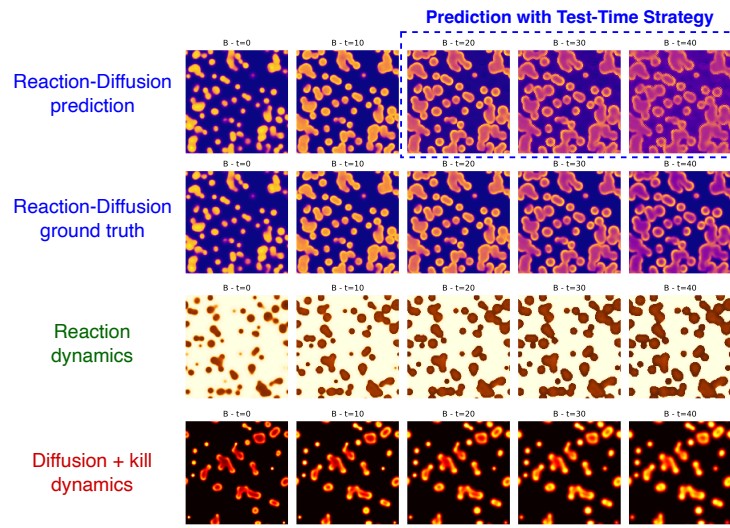

Figure 2: **Test-time generalization on Gray-Scott equations.** Our neural operator search correctly predicts an unseen, non-trivial dynamics (compare first and second rows), which differs substantially from the pure reaction (third row) or pure diffusion (fourth row) seen during training, demonstrating that our method, based on combining simple operators, can capture complex phenomena.

250 temporal snapshots on a 256-point spatial grid over $T = 4$ seconds and a periodic domain of length $l = 16$. We employ the solver from (Brandstetter et al., 2022) to generate the trajectories.

**2D Reaction-Diffusion.** Our most challenging benchmark is the Gray-Scott reaction-diffusion equation from The Well (Ohana et al., 2024):

$$\frac{\partial A}{\partial t} = D_A \nabla^2 A - \delta A B^2 + F(1 - A) \,,$$

$$\frac{\partial B}{\partial t} = D_B \nabla^2 B + \delta A B^2 - (F + k)B \,.$$

This system models the spatiotemporal evolution of two chemical species parameterized by diffusion coefficients $D_A, D_B$, reaction strength $\delta$, feed rate $F$ for species $A$, and kill rate $k$ for species $B$. We construct training data using two operator types: (1) diffusion-kill operators with fixed diffusion coefficients $D_A = 2 \times 10^{-5}$, $D_B = 1 \times 10^{-5}$, disabled reaction ($\delta = 0$, $F = 0$), and kill rates $k$ spanning 20 values in $\{0.051, 0.052, 0.053, \ldots, 0.069, 0.070\}$; (2) pure reaction operators with disabled diffusion ($D_A = D_B = 0$), unit reaction strength ($\delta = 1$), zero kill rate ($k = 0$), and feed rates $F$ taking 20 values in $\{5, 10, \ldots, 95, 100\} \times 10^{-3}$. The spatial domain employs a $128 \times 128$ grid with periodic boundary conditions. We generate 512 trajectories per parameter configuration, using clustered gaussians as initial conditions, simulating 50 seconds and retaining 50 temporal snapshots. We employ the solver from Ohana et al. (2024) to generate the dataset.

**Implementation.** We use the following hyperparameters for our test-time search strategies. **Uniform:** $T = 100$ combinations for advection-diffusion and combined equation, $T = 200$ for Gray-Scott, with maximum composition length $M = 3$. **Beam:** We subsample $N = 256$ operators (advection-diffusion), $N = 96$ operators (combined equation), and $N = 40$ operators (Gray-Scott). We use beam width $B = 4$ for advection-diffusion and combined equation, $B = 8$ for Gray-Scott, maximum composition length $M = 5$, and improvement threshold of 5%.

**Baselines.** We compare against state-of-the-art approaches across different methodological categories. All methods are trained from scratch on the same training datasets designed for this study. We use the next-step prediction as the learning objective. **DISCO (Original)** (Morel et al., 2025): We validate that our framework systematically improves upon the original DISCO approach, which performs predictions by encoding out-of-distribution trajectories and directly predicting dynamics

without test-time adaptation. **MPP** (McCabe et al., 2023): We compare against the Axial Vision Transformer (Ho et al., 2019) architecture designed for multiple physics pretraining, representing current state-of-the-art performance in large-scale physics foundation models. **Zebra** (Serrano et al., 2025): We include this autoregressive transformer inspired by language modeling. While primarily designed for one-shot and few-shot adaptation, Zebra provides a valuable comparison as a generative model that requires significantly higher computational resources than MPP. **GEPS** (Koupaï et al., 2024): We compare against this meta-learning framework designed for efficient few-shot adaptation to changing dynamics. We train it using an environment-based perspective (Yin et al., 2022; Kirchmeyer et al., 2022). This method employs LoRA-based adaptation (Hu et al., 2021), making it an excellent comparison point for efficient weight fine-tuning approaches.

**Test Evaluation.** We evaluate all methods by unrolling predictions over $H = 34$ steps for the advection-diffusion equation, $H = 50$ steps for the combined equation, and $H = 32$ steps for Gray-Scott. All experiments use a history of $L = 16$ snapshots as context, either for direct prediction (MPP, Zebra, original DISCO) or for adaptation (GEPS and our framework). We report the average NRMSE over the entire predicted trajectory as the primary evaluation metric.

## 5.2 PARAMETER EXTRAPOLATION

**Setting.** In this section, we investigate extrapolation capabilities on advection-diffusion systems by testing higher advection speeds $c \in [1, 3]$ and higher diffusion coefficients $D \in [1, 3]$. While higher advection speeds present significant challenges for classical numerical solvers due to transport dominance, higher diffusion coefficients generally provide better numerical stability through smoothing effects. We also examine extrapolation performance for the nonlinear advection term $\alpha \in [1, 2]$ and dispersion coefficient $\gamma \in [1, 2]$ in the combined equation.

**Results.** Table 1 shows that our test-time operator composition strategy consistently improves upon the original DISCO by orders of magnitude across all benchmarks, demonstrating that neural operator splitting can extract substantially greater generalization capabilities from pretrained models compared to direct out-of-distribution encoding. The beam search variant achieves the strongest performance, with improvements ranging from 11× on advection speed extrapolation to over 200× on diffusion coefficient tasks. Among the baselines, GEPS shows competitive performance on nonlinear advection and reasonable results on diffusion tasks, but exhibits instability on dispersion extrapolation. We observed that GEPS fine-tuning often leads to diverging operators during rollout in out-of-distribution settings, consistent with previous findings on gradient-based adaptation methods (Serrano et al., 2025). Zebra particularly struggles on advection-diffusion tasks, which we attribute to limitations in discrete tokenization for capturing the high-frequency dynamics present in our fractal-based initial conditions. MPP provides robust but modest performance across tasks, never excelling in these extrapolation scenarios.

Table 1: **Zero-shot performance on PDE parameter extrapolation.** Average NRMSE over $H$ predicted steps (lower is better) on PDEs with coefficients outside the training range. With fixed weights (no finetuning), standard models struggle to generalize, whereas applying our adaptive operator splitting method to DISCO yields drastic improvements without retraining.

| Method | Advection Diffusion | | Combined Equation | |
|---|---|---|---|---|
| | Adv. speed $c$ | Diffusion $D$ | Nonlin. Adv. $\alpha$ | Dispersion $\gamma$ |
| MPP | 0.588 | 0.409 | 0.134 | 0.369 |
| Zebra | 1.07 | 1.579 | 0.128 | 0.448 |
| GEPS | 0.848 | 0.267 | 0.0203 | 0.782 |
| DISCO (Original) | 0.768 | 0.159 | 0.0887 | 1.007 |
| **Ours (Uniform)** | 0.113 | 0.0546 | 0.0266 | 0.0700 |
| **Ours (Beam)** | **0.0517** | **0.002** | **0.0159** | **0.0215** |

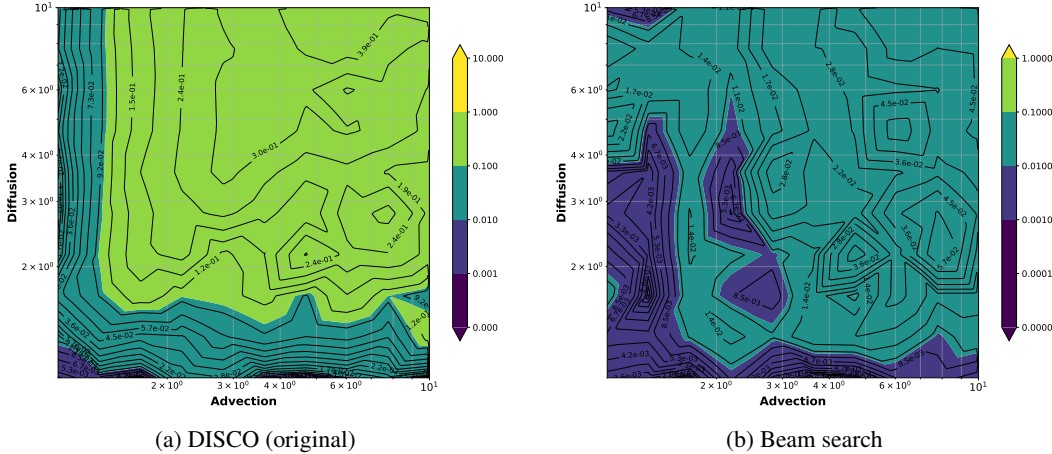

(a) DISCO (original)

(b) Beam search

Figure 3: **Zero-shot performance on advection-diffusion combinations. (a):** Single forward pass of the pretrained model. **(b):** Our test-time method with operators obtained via beam search. Error is the NRMSE averaged over 34 steps.

## 5.3 PHYSICS COMPOSITION

**Setting.** We evaluate the compositional capabilities of neural operators by testing their ability to combine previously isolated physical processes. For the advection-diffusion system, test cases combine both mechanisms with coefficients sampled uniformly from $c \in [0, 1]$ and $D \in [0, 1]$, representing dynamics never seen during training where only individual processes were present.

For the combined equation dataset, we test four types of multi-physics compositions: (1) nonlinear advection + diffusion with $\alpha \in [0, 1], \beta \in [0, 0.4]$; (2) nonlinear advection + dispersion with $\alpha \in [0, 1], \gamma \in [0, 1]$; (3) diffusion + dispersion with $\beta \in [0, 0.4], \gamma \in [0, 1]$; and (4) all three processes combined with $\alpha \in [0, 1], \beta \in [0, 0.4], \gamma \in [0, 1]$. Each test case represents a novel composition of operators that were learned in isolation during training.

For the Gray-Scott system, we evaluate compositional generalization using 1,600 test trajectories spanning the full parameter space defined by the Cartesian product of feed rates $F$ and kill rates $k$ from the training distribution, but combining reaction and diffusion processes that were separated during training.

Table 2: **Zero-shot generalization to unseen PDE combinations.** Average NRMSE over $H$ predicted steps (lower is better) on unseen combinations of physical phenomena (i.e., operators). During training, models see each phenomenon individually (e.g., pure nonlinear advection). At test time, multiple phenomena appear simultaneously (e.g., nonlinear advection + dispersion). With fixed weights (no finetuning), standard models struggle to generalize, whereas DISCO with adaptive operator splitting achieves substantial gains without retraining.

| Method | Adv.+Diff. | Nonlin.Adv.+Diff. | Nonlin.Adv.+Disp. | Diff.+Disp. | All Three | React.+Diff. |
|---|---|---|---|---|---|---|
| MPP | 0.270 | 0.050 | 0.105 | 0.0914 | 0.128 | 0.191 |
| Zebra | 0.89323 | **0.0223** | 0.241 | 0.069 | 0.193 | 0.127 |
| GEPS | 0.0392 | 0.0389 | 0.249 | 0.229 | 0.265 | 0.128 |
| DISCO (Original) | 0.170 | 0.085 | 0.100 | 0.120 | 0.164 | 0.245 |
| **Ours (Uniform)** | **0.043** | 0.068 | **0.103** | **0.043** | **0.0753** | **0.0898** |
| **Ours (Beam)** | **0.0150** | 0.0565 | **0.0489** | **0.007** | **0.0364** | **0.0889** |

**Results.** Table 2 shows that our method achieves the best performance on 5 out of 6 composition tasks, with significant improvements over the original DISCO approach. For instance, on diffusion + dispersion, our beam method outperforms DISCO by over 17×, while on advection + diffusion, we achieve a 10× improvement. The beam search variant consistently outperforms uniform sam-

pling, indicating that efficient search over operator compositions is crucial for identifying effective decompositions of new dynamics.

Among baselines, we observe distinct performance patterns. Zebra achieves the strongest performance on nonlinear advection + diffusion, while our method still improves upon DISCO. GEPS demonstrates competitive performance on combinations like advection + diffusion but struggles with more complex compositions involving dispersion terms, often exhibiting rollout instabilities after finetuning. MPP provides consistent but modest performance across all tasks.

Figure **??** illustrate these quantitative results through qualitative rollout predictions. For the Gray-Scott system, our method successfully predicts the complex reaction-diffusion patterns that emerge from our operator composition, closely matching the ground truth dynamics. Similarly, in Figure 3 for advection-diffusion combinations, our approach better recovers the combination of transport and diffusive effects compared to the original DISCO.

**Test-time scaling laws.** Figure 4 shows that as we increase the number of trials, both fitting error and prediction error consistently decrease following power-law-like decay, with improved fit error directly correlating with better prediction accuracy. Beyond prediction accuracy, our method enables interpretable parameter identification by inspecting the selected operator combinations and summing their true underlying PDE coefficients. The right panel shows that as predictive accuracy improves with more computation, parameter estimation accuracy for both advection speed and diffusion coefficient also improves systematically.

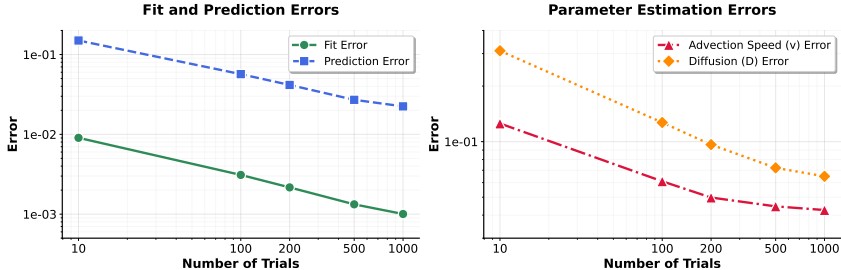

Figure 4: **Test-time scaling laws.** (Left) Performance of our adaptive operator splitting method measured as an average NRMSE over 34 predicted steps (lower is better), as a function of the compute at test-time. (Right) Mean Absolute Error (MAE) for PDE parameter identification as a function of the number of optimization trials, showing convergence to accurate parameter estimates with increased sampling.

## 6 CONCLUSION

We introduce a method for predicting unknown dynamical systems governed by unseen PDEs at test time, without modifying the model weights. Our approach builds on a pretrained DISCO model (Morel et al., 2025), which identifies a collection of operators from the training data. At test time, it searches for the best combination of these operators, using operator splitting, to fit the observed out-of-distribution trajectory. While standard surrogate models struggle with zero-shot generalization, our method achieves state-of-the-art performance, approaching the accuracy of models operating in-distribution.

The main limitation of our approach lies in the operator splitting itself, which requires composing operators with matching input and output domains. Future work will investigate generalization across dimensionality (e.g., 2D to 3D), spatial domains (e.g., different grid structures), and physical fields (e.g., velocity, momentum, density).

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

# A DATASET DETAILS

## A.1 ADVECTION-DIFFUSION

We generate synthetic trajectories for the 1D advection-diffusion equation

$$\frac{\partial u}{\partial t} + v\frac{\partial u}{\partial x} = D\frac{\partial^2 u}{\partial x^2} \tag{1}$$

with periodic boundary conditions, where $v$ is the advection speed and $D$ is the diffusion coefficient. The dataset uses analytical solutions computed via Fourier spectral methods to avoid numerical errors.

**Physical Parameters.** During training, we generate 50% pure advection cases ($v \sim$ Uniform$(0.01, 1.0)$, $D = 0$) and 50% pure diffusion cases ($v = 0$, $D \sim$ Uniform$(0.001, 1.0)$).

**Initial Conditions.** We generate complex initial conditions using Fractaloid with random phase patterns, which create self-similar signals with power-law spectra. These patterns are constructed as trigonometric polynomials

$$u_0(x) = \sum_{k=1}^{\text{degree}} a_k k^{-\text{power}} \sin(k\theta + \phi_k), \tag{2}$$

where $a_k$ are independent Gaussian coefficients and $\phi_k$ are random phases. We use degree $= 256$ and power is sampled uniformly in $[1, 4]$, then normalize each initial condition to zero mean and unit variance. For testing, we fix the power to 3.

**Analytical Solutions.** We compute exact solutions using Fourier spectral methods. In spectral space, the solution evolves as $\hat{u}(k, t) = \hat{u}_0(k) \exp(-Dk^2 t) \exp(-ikvt)$, which we transform back to physical space via inverse FFT. The spatial domain has length $L = 16.0$ with $n_x = 256$ grid points, evolved over $n_t = 100$ time steps to final time $T = 10.0$.

## A.2  COMBINED-EQUATION

We follow the dataset generation approach of Brandstetter et al. (2022), with key distinctions in the physics formulation for training data generation and the exclusion of forcing terms. The combined equation is governed by the following PDE:

$$\partial_t u + \partial_x(\alpha u^2 - \beta \partial_x u + \gamma \partial_{xx} u) = 0, \tag{3}$$

subject to periodic boundary conditions and initial conditions

$$u_0(x) = \sum_{j=1}^{J} A_j \sin(2\pi \ell_j x / l + \phi_j). \tag{4}$$

This formulation combines three fundamental physical mechanisms: nonlinear advection ($\alpha u^2$), linear diffusion ($-\beta \partial_x u$), and dispersion ($\gamma \partial_{xx} u$). For each initial condition, we sample the Fourier mode coefficients: $A_j \sim \text{Uniform}([-0.5, 0.5])$, $\ell_j \sim \text{Uniform}(\{1, 2, 3, 4, 5\})$, and $\phi_j \sim \text{Uniform}([0, 2\pi])$ with $J = 5$ modes.

**Training dataset** The training data is generated using parameter combinations $(\alpha, 0, 0)$, $(0, \beta, 0)$, and $(0, 0, \gamma)$. The coefficients are sampled uniformly from $\alpha \in [0, 1]$, $\beta \in [0, 0.4]$, and $\gamma \in [0, 1]$. We generate 8,192 training trajectories for each isolated physics across 128 parameter configurations. Each trajectory contains 250 temporal snapshots on a 256-point spatial grid over $T = 4$ seconds with a periodic domain of length $l = 16$.

## A.3  REACTION-DIFFUSION

Our most challenging benchmark is the Gray-Scott reaction-diffusion system from The Well (Ohana et al., 2024):

$$\frac{\partial A}{\partial t} = D_A \nabla^2 A - \delta AB^2 + F(1 - A), \tag{5}$$

$$\frac{\partial B}{\partial t} = D_B \nabla^2 B + \delta AB^2 - (F + k)B. \tag{6}$$

This system models the spatiotemporal evolution of two chemical species parameterized by diffusion coefficients $D_A, D_B$, reaction strength $\delta$, feed rate $F$ for species $A$, and kill rate $k$ for species $B$.

**Training Data Generation.** We construct training data using two types of operator. First, *diffusion-kill operators* use fixed diffusion coefficients $D_A = 2 \times 10^{-5}$, $D_B = 1 \times 10^{-5}$, disabled reaction terms ($\delta = 0$, $F = 0$), and kill rates $k$ spanning 20 values in $\{0.051, 0.052, \ldots, 0.070\}$. Second, *pure reaction operators* disable diffusion ($D_A = D_B = 0$), set unit reaction strength ($\delta = 1$), zero kill rate ($k = 0$), and vary feed rates $F$ across 20 values in $\{5, 10, \ldots, 100\} \times 10^{-3}$.

The spatial domain employs a $128 \times 128$ grid with periodic boundary conditions. We generate 512 trajectories per parameter configuration, simulating 50 seconds and retaining 50 temporal snapshots using the solver from Ohana et al. (2024).

**Initial Conditions.** To ensure fair evaluation of dynamics identification and extrapolation capabilities, we address the distinct field characteristics produced by reaction versus diffusion dynamics. We begin with clustered Gaussian initial conditions, then evolve them for a random duration between 0 and 100 seconds using the full reaction-diffusion dynamics. The resulting evolved states serve as initial conditions for generating the isolated reaction and diffusion training trajectories. This procedure mitigates potential frequency bias across all methods and enables the assessment of operator learning rather than initial condition adaptation.

# B  IMPLEMENTATION DETAILS

## B.1  DISCO IMPLEMENTATION

**Hyperparameters** We use the recommended default configuration from Morel et al. (2025) with targeted modifications for our experiments. For the transformer encoder, we employ a hidden dimension of 128, patch sizes of 8 in 1D and $8 \times 8$ in 2D, and 4 encoder blocks with 4 attention

heads each using relative position bias. We introduce a bottleneck projection layer that reduces the 128-dimensional transformer output to $C$ channels, where $C = \{2, 3, 2\}$ for advection-diffusion, combined-equation, and reaction-diffusion respectively. This bottleneck layer is inserted before DISCO's original MLP decoder, representing a minimal architectural change that improves generalization across initial conditions.

For the neural ODE component, we select the RK4 solver with problem-specific integration time spans: $dt = \{0.1, 0.016, 0.016\}$ for advection-diffusion, combined-equation, and reaction-diffusion respectively. We apply periodic boundary conditions and configure the operator network with 8 base channels and a $2\times$ bottleneck multiplier for efficient ODE parameter prediction from transformer representations.

We train models for 300,000 iterations on advection-diffusion and combined-equation tasks, and 100,000 iterations for reaction-diffusion. We use AdamW optimizer with a base learning rate of $3 \times 10^{-4}$, cosine annealing scheduler, and weight decay of $1 \times 10^{-4}$.

## B.2 TRAINING RECIPE

The original DISCO training procedure uses the operator to predict the frame immediately following the input encoder sequence. We found that increasing input diversity to the operator network produces more robust operators that generalize across different initial conditions.

We therefore propose an alternative training strategy based on contextual learning. For advection-diffusion equations, we sample two trajectories that follow identical dynamics: we encode trajectory 1 with the hypernetwork to obtain an operator, then apply this operator to predict the next timestep of trajectory 2. This in-context approach draws inspiration from Serrano et al. (2025).

For Combined-equation and Gray-Scott systems, we adopt an environment-based training paradigm to enable fair comparison with GEPS Koupaï et al. (2024). We assume knowledge of which trajectories belong to the same environment and implement a codebook updated via exponential moving average following Oord et al. (2017). During training, we randomly select either the encoder-derived code (50% probability) or the corresponding environment code from the codebook (50% probability), ensuring the encoder learns meaningful representations while maintaining environment consistency.

---

**Algorithm 2** Random Operator Composition Search

**Require:** Test trajectory $u_{\text{test}}^{1:L}$, operator dictionary $\{f_1, \ldots, f_N\}$, number of trials $N_{\text{trials}}$, maximum composition length $M$
**Ensure:** Best operator subset $S^*$
1: Initialize best operator subset $S^* = \{\arg\min_{f_i \in \{f_1, \ldots, f_N\}} \mathcal{L}(\{f_i\})\}$
2: Initialize best loss $\mathcal{L}^* = \mathcal{L}(S^*)$
3: **for** $i = 1$ to $N_{\text{trials}}$ **do**
4:     Sample composition length $m \sim \text{Uniform}(\{1, 2, \ldots, M\})$
5:     Sample operator subset $S_i \sim \text{Uniform}(\text{subsets of } \{f_1, \ldots, f_N\} \text{ with size } m)$
6:     Compute loss $\mathcal{L}_i = \mathcal{L}(S_i)$
7:     **if** $\mathcal{L}_i < \mathcal{L}^*$ **then**
8:         $S^* = S_i$
9:         $\mathcal{L}^* = \mathcal{L}_i$
10:     **end if**
11: **end for**
12: **return** $S^*$

---

## B.3 BASELINES

**MPP** We use the recommended default hyperparameters with periodic boundary conditions and 6 encoder blocks, employing a hidden dimension of 384 for 2D experiments, and train for 100,000 iterations using AdamW optimizer with a learning rate of $5 \times 10^{-4}$ and batch size of 64.

**Zebra** We use recommended defaults, and used respectively 64, 32, 256 tokens to encode each frame in advection-diffusion, combined-equaiton, reaction-diffusion. We train the model without in-context example, using a maximum history length of 50 frames, 66, and 32 frames respectively for advection, combined equation, reaction-diffusion. We use random sampling with a temperature of 0.1 at inference.

**Zebra** We adopt the recommended default configuration, using 64, 32, and 256 tokens respectively to encode each frame for advection-diffusion, combined-equation, and reaction-diffusion tasks. We train without in-context examples, employing maximum history lengths of 50, 66, and 32 frames for advection-diffusion, combined-equation, and reaction-diffusion respectively. At inference, we use random sampling with a temperature of 0.1.

**GEPS** We use the CNN1D and CNN2D implementations from the original codebase, training for 100,000 steps with AdamW optimizer and cosine learning rate scheduling. Since GEPS requires environment information during training, we provide labels indicating which trajectories belong to the same environment. At inference, we address rollout instabilities by performing multiple optimization runs (100, 500, and 2000 gradient steps) and report the best test set performance across these attempts.

## C  QUALITATIVE RESULTS

**Combined equation** We can see in Figure 5 , 6, 7, 8 that our test-time operator splitting strategy demonstrates remarkable capability in matching ground truth dynamics over extended rollouts, despite operating out-of-distribution and being trained solely for single-step prediction. The model maintains high fidelity predictions throughout the majority of the 100-step rollout, with some error accumulation becoming visible after approximately 70 timesteps, which is expected for such long-horizon extrapolation tasks.

**Reaction diffusion** We provide an augmented comparison of the dynamics seen during training with the second channel in Figure 9. We also show additional comparisons of predictions and ground truths in Figure 10, 11, 12.

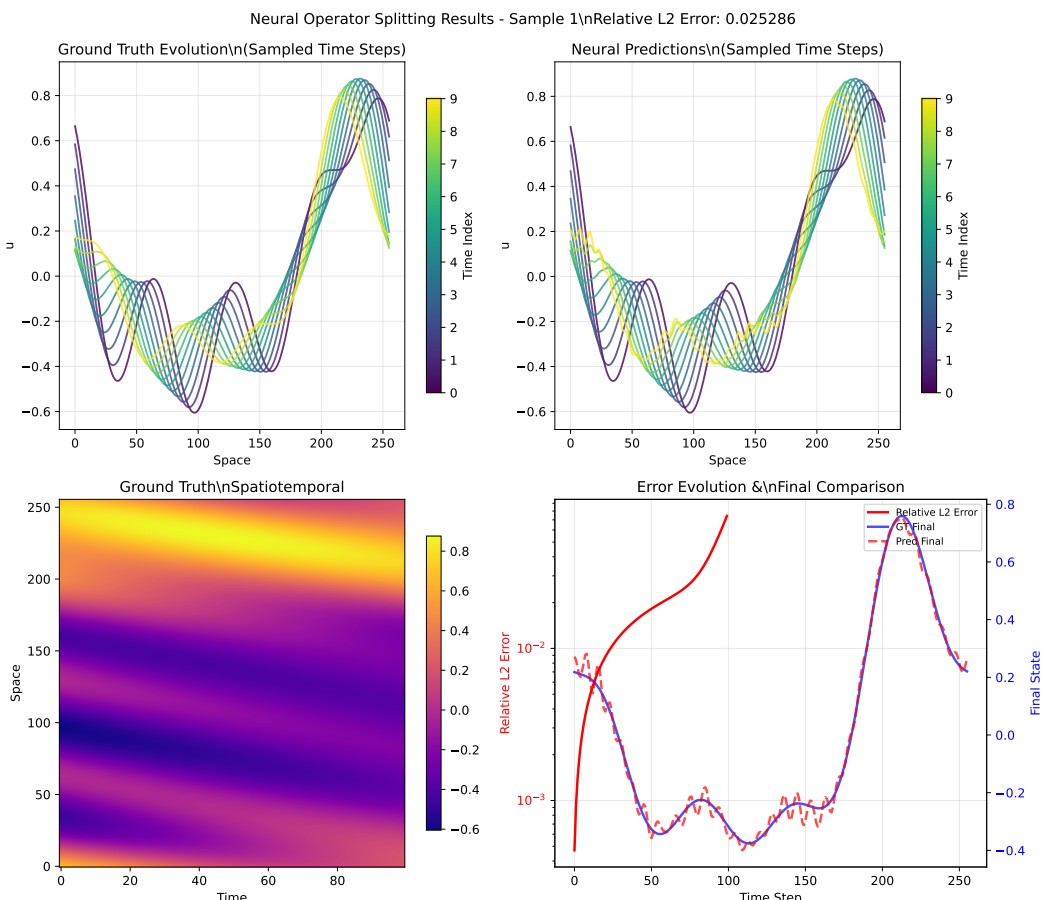

Figure 5: **OOD trajectory prediction on diffusion+dispersion equations.** We select operators using beam search and autoregressively unroll dynamics for 100 timesteps. The top panels show ground truth (left) and model predictions (right). The bottom panel displays error evolution throughout the rollout and compares the final prediction against ground truth.

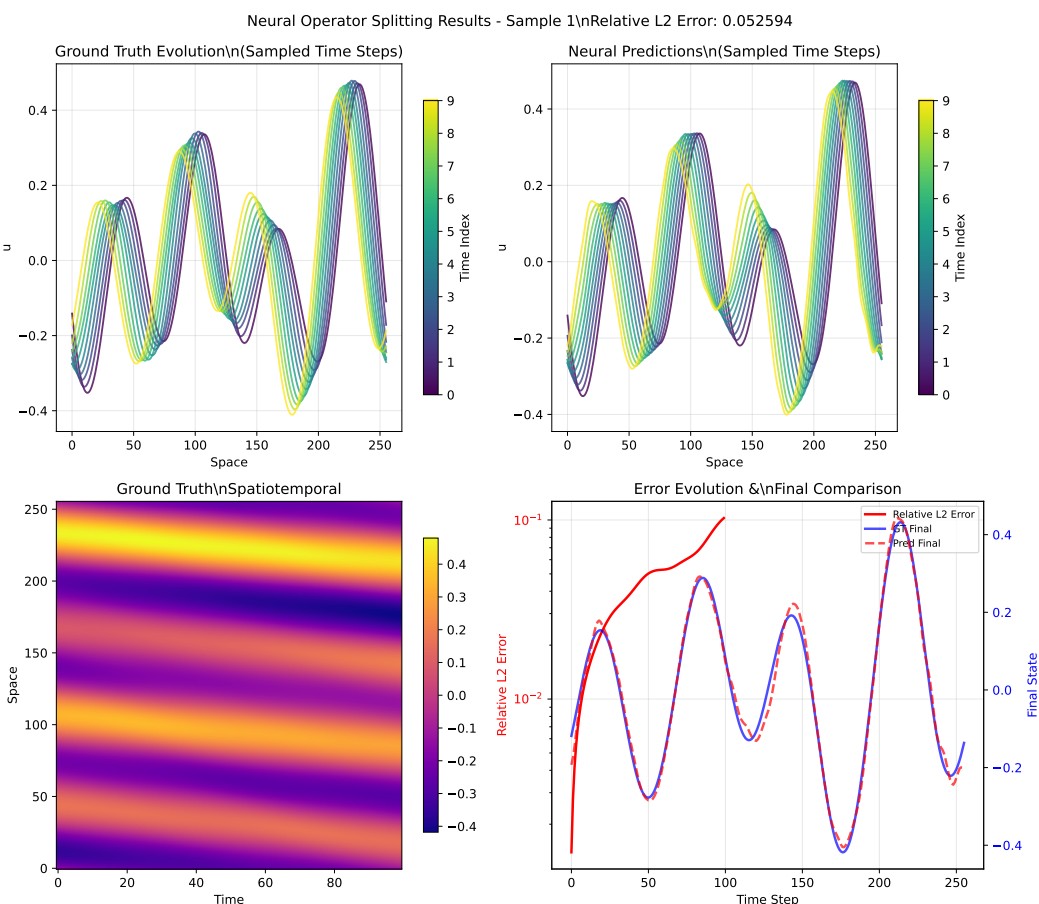

Figure 6: **OOD trajectory prediction on nonlinear advection+dispersion.** We select operators using beam search and autoregressively unroll dynamics for 100 timesteps. The top panels show ground truth (left) and model predictions (right). The bottom panel displays error evolution throughout the rollout and compares the final prediction against ground truth.

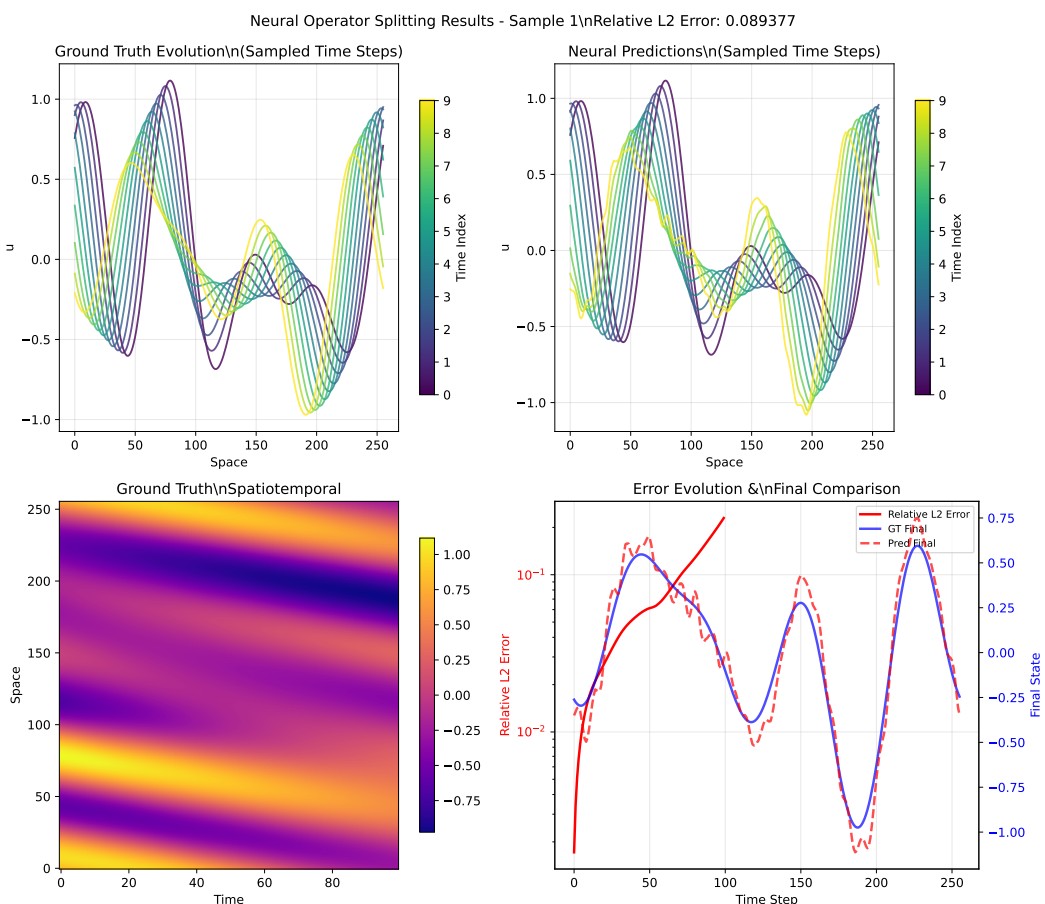

Figure 7: **OOD trajectory prediction on nonlinear advection+dispersion+diffusion.** We select operators using beam search and autoregressively unroll dynamics for 100 timesteps. The top panels show ground truth (left) and model predictions (right). The bottom panel displays error evolution throughout the rollout and compares the final prediction against ground truth.

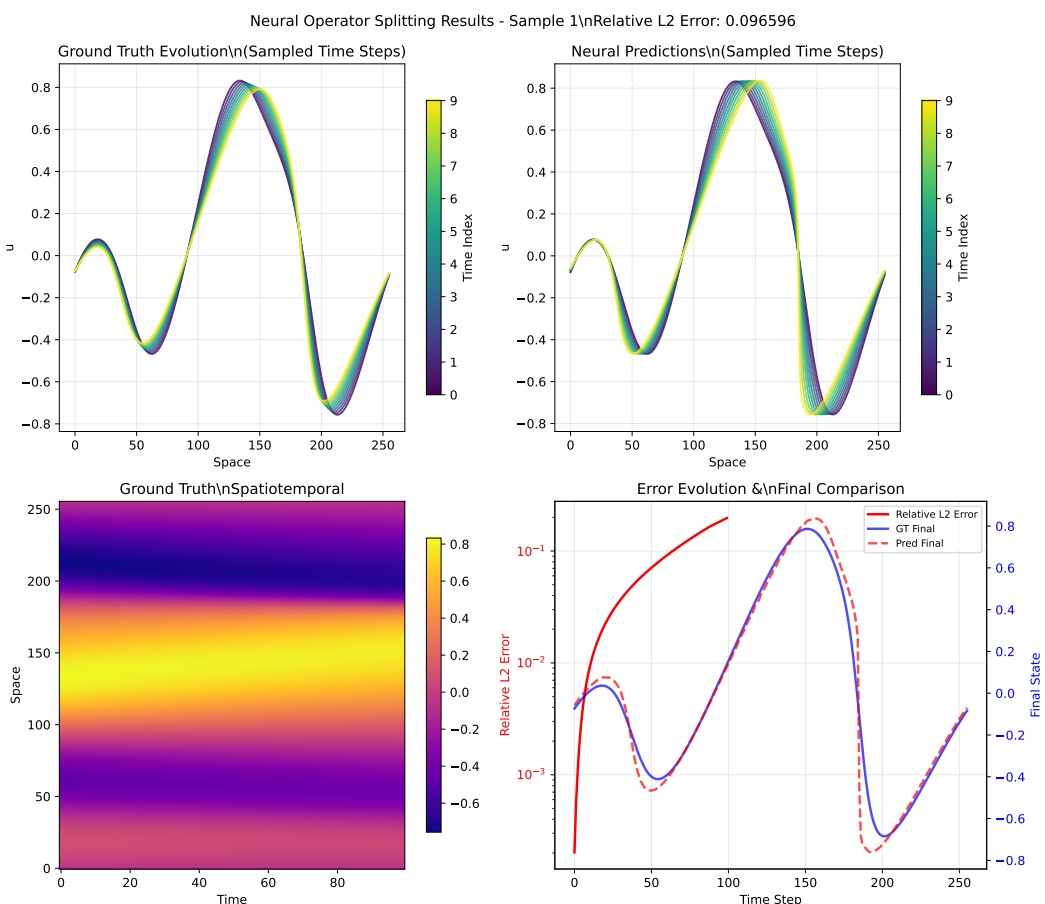

Figure 8: **OOD trajectory prediction on nonlinear advection+diffusion.** We select operators using beam search and autoregressively unroll dynamics for 100 timesteps. The top panels show ground truth (left) and model predictions (right). The bottom panel displays error evolution throughout the rollout and compares the final prediction against ground truth.

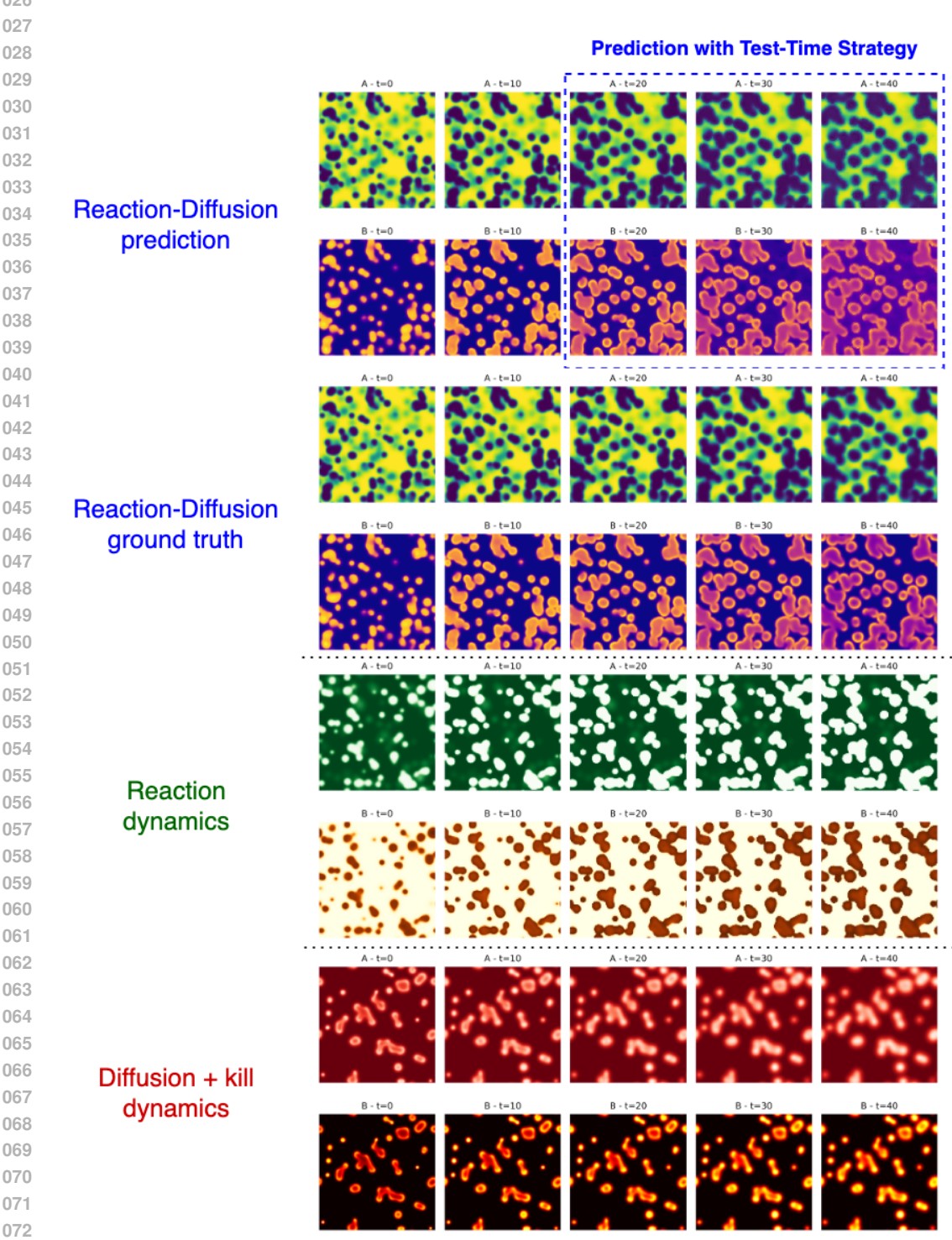

Figure 9: **OOD trajectory prediction on Gray-Scott equations.** Visualization of operator splitting decomposition for Gray-Scott reaction-diffusion dynamics. The top section compares our test-time strategy predictions (blue box) against ground truth for the full reaction-diffusion system, showing species A (yellow-green) and B (red-blue) concentrations. The bottom section displays the kind of dynamics seen during training: pure reaction terms (green/brown) and diffusion with kill terms (red/orange) for both species.

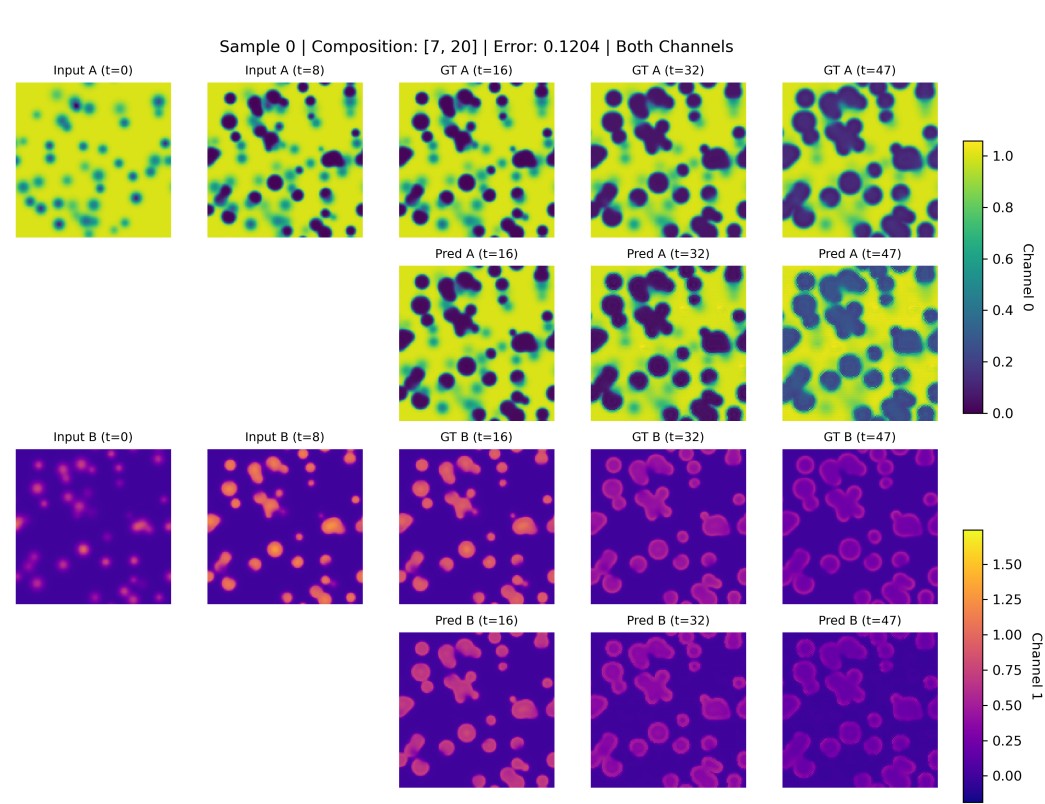

Figure 10: **OOD trajectory prediction on Gray-Scott equations.** The first two rows show ground truth (top) and predicted (second) concentrations for species $A$. The bottom two rows display ground truth (third) and predicted (bottom) concentrations for species $B$.

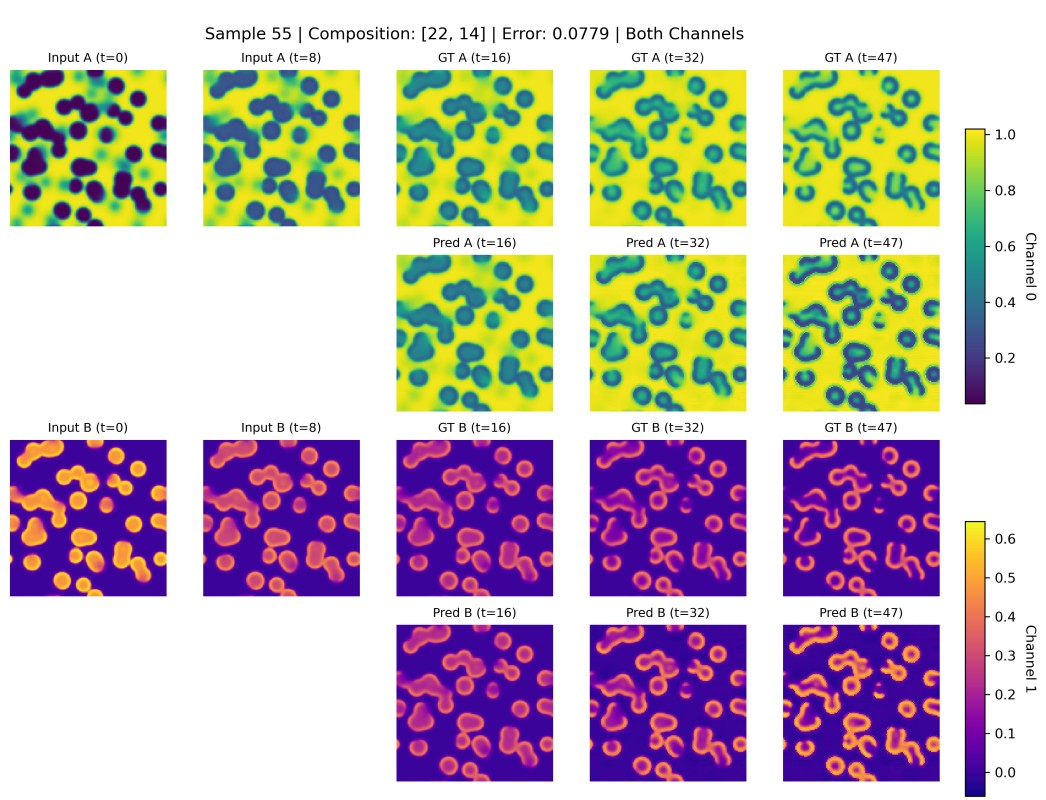

Figure 11: **OOD trajectory prediction on Gray-Scott equations.** The first two rows show ground truth (top) and predicted (second) concentrations for species $A$. The bottom two rows display ground truth (third) and predicted (bottom) concentrations for species $B$.

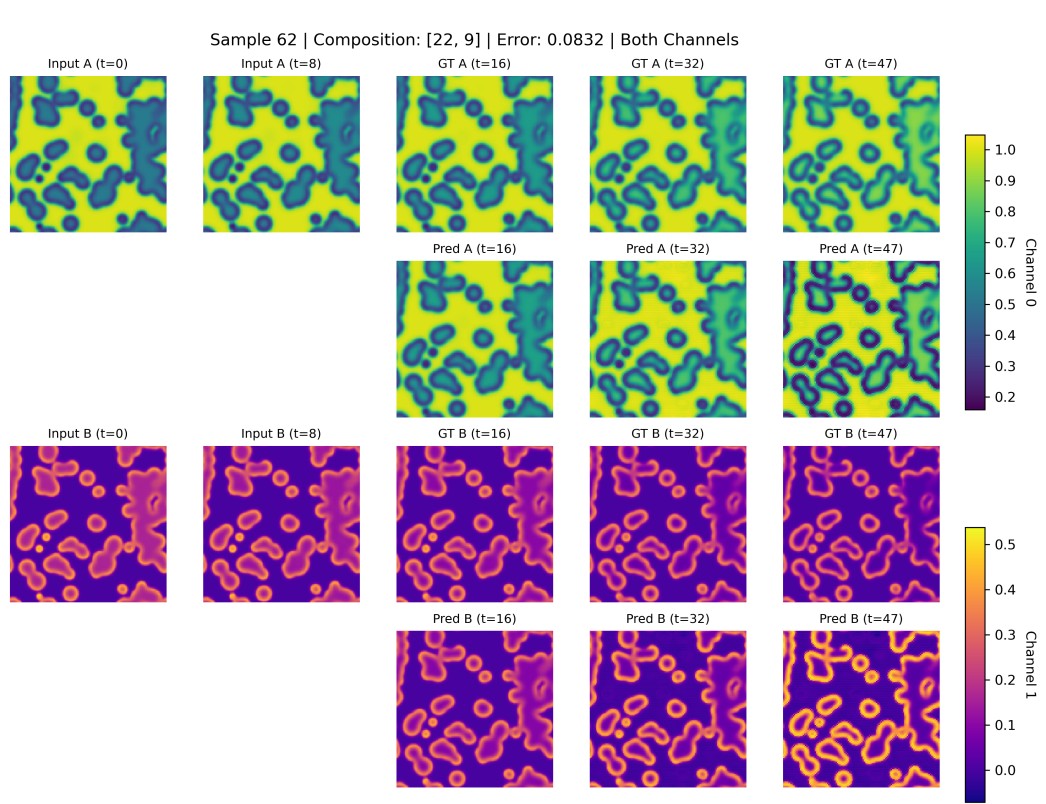

Figure 12: **OOD trajectory prediction on Gray-Scott equations.** The first two rows show ground truth (top) and predicted (second) concentrations for species $A$. The bottom two rows display ground truth (third) and predicted (bottom) concentrations for species $B$.

