# OpenReview forum: "Test-time Generalization for Physics through Neural Operator Splitting"
_ICLR.cc/2026/Conference — Submitted to ICLR 2026_

### Official Review · Reviewer_2BLA · 2025-10-30

**Soundness:** 2
**Presentation:** 3
**Contribution:** 3
**Rating:** 4
**Confidence:** 4

**Summary:**

This paper presents a test-time adaptation approach for neural surrogate models to improve the out-of-distribution (OOD) performance. The authors combine pretrained operators and a beam search method, and a test-time scaling law is provided. There are two OOD scenarios considered in this paper, i.e., parameter extrapolation and operator composition. Several numerical experiments have shown the superiority of the proposed test-time computation framework compared to baseline models.

Contributions:

- The authors are tackling the critical OOD problem in scientific machine learning.

- The test-time adaptation in scientific machine learning is new.

**Strengths:**

- The test-time generalization in scientific ML is under-explored. This paper investigates a critical topic.

- This paper is well-written. The motivation and formulation of test-time scaling are well presented.

**Weaknesses:**

This is a good topic, but I have a few concerns regarding the experiments part.

- First, I think the OOD scenarios can be broader. Apart from the parameter extrapolation and operator composition, the authors might also consider unseen initial conditionals, boundary conditions, geometries, etc. Please refer to the unisolver paper [1].

- Second, it would be good to have a more explicit discussion of computational overhead. On Page 2, the authors also claimed that “This test-time strategy comes at a higher computational cost, but enables to better adapt when faced with unseen dynamics.” It is good to see the performance improvement in the paper, but it would also be useful to see how much extra computation this actually requires. I think including some numbers or plots on runtime or resource use would give readers a better sense of the tradeoff between performance and efficiency.

- Third, the tested PDEs can be broader. The Navier-Stokes equations seem to be one of the standard benchmark datasets that people will test. The authors might also consider testing on more diverse PDEs.

---

**Refs:**

[1] Zhou, Hang, et al. "Unisolver: PDE-Conditional Transformers Towards Universal Neural PDE Solvers." Forty-second International Conference on Machine Learning.

**Questions:**

- Could you clarify the motivation for using LoRA? How large is your model? If it is relatively small, why was full model fine-tuning not considered as an alternative?

- How does this method scale to 3D PDEs?

- In the Abstract, it is better to cite the DISCO paper there to avoid confusion.

- On Page 9, line 440, there is a citation issue for Figure ??.

---

> ### Author Response · Authors · 2025-11-24
> **Response**
>
> Thank you for your clear and actionable comments, which guided us to broaden the evaluation and improve the compute analysis of our method.
> ### Weakness 1: Limited OOD scenarios
> We focus on dynamics OOD (parameter shifts and operator composition) which are especially challenging for neural surrogates. Geometry/BC OOD is orthogonal and best handled by specialized geometric operators. To broaden the evaluation, we added Experiment A, which includes another complex PDE system (Navier–Stokes 2D).
> ### Weakness 2: Missing computational overhead analysis
> Experiment B provides a detailed compute analysis. Our method operates in a favorable regime where increased compute consistently improves performance, unlike the baselines. This clarifies the overhead–accuracy tradeoff.
> ### Weakness 3: Limited diversity of PDEs
> Experiment A expands the evaluation to Navier–Stokes via Euler–Diffusion composition, demonstrating applicability to other challenging PDEs.
> ### Question: Motivation for LoRA
> LoRA is used only by the GEPS baseline. Our method performs no fine-tuning and relies solely on operator search and composition. We will clarify this distinction.
> ### Question: Scalability to 3D PDEs
> Search and splitting are dimension-agnostic. The cost increases only in proportion to the operator evaluations. We will add a short clarification in the discussion.

---

### Official Review · Reviewer_T1WF · 2025-10-31

**Soundness:** 3
**Presentation:** 2
**Contribution:** 3
**Rating:** 6
**Confidence:** 4

**Summary:**

This paper introduces a test-time adaptation strategy for neural PDE surrogates that enables zero-shot generalization to out-of-distribution dynamics. The method builds on DISCO, extracting a dictionary of neural operators from training trajectories, then uses beam search with operator splitting at test time to compose these operators and approximate unseen dynamics. The approach is evaluated on parameter extrapolation and physics composition tasks across three benchmarks. Results demonstrate significant improvements over baselines.

**Strengths:**

- The core idea of combining a learned dictionary of neural operators with classical operator splitting at test time is, to my knowledge, highly novel. This is a well-motivated approach of bringing forth test-time computation to the realm of PDE surrogate modeling, drawing clever parallels to LLM inference techniques (beam search, best-of-N sampling).
- A valuable feature of this formulation is the interpretability aspect. By analyzing the selected operator combinations, we can perform zero-shot parameter estimation.
- The benchmarks performed demonstrate strong performance achieving order of magnitude improvements over the baselines considered.

**Weaknesses:**

- While operator splitting has theoretical foundations for classical numerical methods, there's no analysis of when/why it works for learned neural operators. What's the role of the approximation error of the individual operators? Have you tried to study the convergence behavior of these splitting schemes (Lie or Strang)? How does the approximation error of the individual neural operators interact with the splitting error of the numerical scheme?
- While computational complexity is stated, actual runtime comparisons with baselines are absent. How does test-time compute compare to simply fine-tuning?
-  Training only on single-operator dynamics seems like a constraint. How would the method perform if training included some operator combinations? Additionally, it seems like the framework is restricted to purely additive composition. Can the method ever isolate a pure diffusion operator if its dictionary only contained reaction-diffusion and reaction operators?

**Questions:**

Please address weaknesses. Additionally:
- While the method is impressive with the reaction-diffusion case that's demonstrated, can you possibly extend this to other cases? I'd like to see this being extended to other problems of interest. If the model were trained on a dictionary containing operators for the Euler equations (inviscid flow) and a separate set of operators for viscous diffusion, could the test-time search successfully discover this composition to approximate solutions to the full Navier-Stokes equations?

---

> ### Author Response · Authors · 2025-11-24
> **Response**
>
> Thank you for your constructive and technically focused remarks, which directly motivated new analyses and experiments.
> ### Weakness 1: Lack of analysis for learned operator splitting
> We addressed this by adding Experiment C, where we track how individual operator errors (heat and dispersion) affect splitting accuracy. The results show that the composed solver is dominated by the least accurate operator, providing clear empirical insight into how splitting behaves with learned components.
> ### Weakness 2: Missing runtime comparisons
> Experiment B provides FLOP-based comparisons between our test-time method, GEPS, and direct inference. We show that our method uses more compute but improves predictably with FLOPs, whereas GEPS plateaus early. This makes the compute–accuracy tradeoff explicit.
> ### Weakness 3: Training only on single-operator dynamics
> The dictionary may include both pure and composite operators (e.g., Gray–Scott). Training pure operators maximizes flexibility for test-time composition. We will clarify this design choice and note that additive composition is a current limitation. Regarding the remark about isolating a single operator, while there exists algebraic constructions which are theoretically possible, they result in operators that are not physically meaningful and thus fall outside the intended scope of our dictionary.
> ### Question: Extension to Euler + Diffusion to obtain Navier–Stokes
> Following your suggestion, we conducted experiments on neural operator splitting of the Navier–Stokes operator into Euler and diffusion components, and we confirm that our approach works as expected. Please refer to the results in Experiment A.

---

### Official Review · Reviewer_jMxQ · 2025-11-01

**Soundness:** 2
**Presentation:** 3
**Contribution:** 2
**Rating:** 2
**Confidence:** 4

**Summary:**

This paper presents an approach for Physics based on test time splitting. The primary idea is to create a library of neural operators using NODE and then combine them during test time.

**Strengths:**

The problem that the paper is trying to solve is relevant and timely, and will have a significant impact. The paper is general is well written.

**Weaknesses:**

Despite the fact that the problem statement is extremely relevant, there are several problems as highlighted below:
(a) The literature review is incomplete. There are works that has previously attempted to solve this problem. For example, ICON (https://www.pnas.org/doi/10.1073/pnas.2310142120), NCWNO (https://www.sciencedirect.com/science/article/pii/S0010465525003844). In fact the idea of combining previously learned solution is something NCWNO has explored previously (although the strategy of combining is slightly different). There also exists Poseidon, which is also in the same space.
(b) As the literature review is incomplete, so is the benchmarking in results section.
(c) The example selected are too simple. Solving such simple problem is not convincing.
(d) The fact that the final operator is a lienar combination of two operators (from dictionary) is somewhat limiting in my opinion.

**Questions:**

a) Why the final operator was considered to be a linear combination of learned operator?
b) In scientific computing, the objective is to predict given the boundary condition and initial condition. While I acknowledge that many previous work has considered time step data as input, this is not that useful as those data will not be available unless a numerical simulator is used. Will the proposed approach work in case we only give initial and boundary condition as input during testing. It seems to me it wont as forming the loss for selecting (through Beam search) the operators from the dictionary will not be possible in such cases.

---

> ### Author Response · Authors · 2025-11-24
> **Response**
>
> Thank you for your detailed and precise feedback, which helped us strengthen both the scope and clarity of our contribution.
> ### Weakness 1: Incomplete literature review
> We will add ICON, NCWNO, and Poseidon to the related work and clarify that they operate in settings different from ours (ODE in-context learning, continual learning with access to training data, or fine-tuning with a single snapshot). In contrast, our method tackles the task of zero-shot test-time adaptation without weight update, which is a different and much more complex setting.
> ### Weakness 2: Incomplete benchmarking
> Because ICON, NCWNO, and Poseidon rely on assumptions incompatible with our setting, we benchmark against DISCO, Zebra, and GEPS, which explicitly target PDE dynamics shifts with trajectory conditioning. We will make this clearer in the manuscript.
> ### Weakness 3: Examples are too simple
> Following your advice, we conducted experiments on a much more complex setting showing that our method is not limited to the examples mentioned in the paper. We also added Experiment A (Euler2D + Diffusion2D composition to obtain Navier–Stokes2D), showing that the splitting also works in this challenging scenario.
> ### Weakness 4: Linear combination limitation
> Thank you for the comment. We agree that purely additive compositions may seem restrictive, but they already cover many practically relevant PDEs. As shown in Experiment A, linear operator composition allows us to recover nontrivial dynamics such as Navier–Stokes 2D by combining the Euler operator with a diffusion term. In this setting, viscosity enters exactly as an additive operator, and it is the key parameter controlling the Reynolds number. Generalizing across different viscosities/Reynolds numbers is an active research challenge, but our results show that even this simple additive framework can express and recover meaningful dynamical variations.
> ### Question: Use with initial/boundary conditions only
> We recognize the importance of the task you mention, but we focused in this paper on generalization to the PDE itself, which is already challenging for neural surrogates.

---

### Author Response · Authors · 2025-11-24
**Global response (1/2)**

We thank all reviewers for their constructive feedback. In response, we conducted three additional experiments that directly address the questions regarding A) applicability to other complex PDE systems, B) computational overhead, and C) splitting behavior. These new results substantively strengthen the paper, and we will integrate them into the revised manuscript where appropriate.


## A) Euler2D + Diffusion2D composition to obtain Navier–Stokes2D
(Addresses: Reviewer jMxQ – simplicity of examples; Reviewer 2BLA – PDE diversity; Reviewer T1WF – extensibility)
To test generalization to more complex PDE systems, we composed:
a neural operator trained only on 2D Euler, and
a neural operator trained only on 2D diffusion.
The composed solver approximates 2D Navier–Stokes without any joint training or fine-tuning.

### Navier–Stokes Approximation with neural operator splitting
| Pretraining Epoch | Euler2D Err | Diff2D Err | Test-time NS Next-Step | Test-time NS Rollout |
|-------------------|-------------|------------|--------------------------|------------------------|
| 10  | 1.42e-02 | 1.38e-03 | 1.26e-02 | 3.17e-01 |
| 30  | 8.99e-03 | 1.27e-04 | 9.08e-03 | 2.46e-01 |
| 50  | 7.66e-03 | 4.07e-05 | 7.10e-03 | 1.84e-01 |
| 80  | 6.86e-03 | 2.23e-05 | 6.53e-03 | 1.76e-01 |
| 100 | 6.55e-03 | 2.10e-05 | 6.42e-03 | 1.73e-01 |

### Conclusion
- As the Euler2D operator improves, the composed Navier–Stokes solver improves consistently without ever training on the full Navier-Stokes dynamics.
 - This experiment shows that additive operator composition alone is expressive enough to recover Navier–Stokes dynamics, indicating that our test-time operator selection and composition framework can, in principle, identify such combinations.

## B) Test-Time Compute Scaling Analysis
(Addresses: Reviewer T1WF – runtime; Reviewer 2BLA – overhead; Reviewer jMxQ – benchmarking clarity)
We performed a detailed compute–accuracy analysis comparing: direct prediction, GEPS fine-tuning, our test-time compute methods: beam search and random sampling.

### Final Compute–Accuracy Comparison
| Method             | FLOPs (B) | Final Error |
|--------------------|-----------|-------------|
| Direct             | 0.34      | 9.99e-02    |
| GEPS Finetuning    | 161.15    | 2.81e-03    |
| Beam Search (Ours) | 612.60    | 4.65e-04    |
| Random (Ours)      | 951.68    | 1.22e-03    |

### Beam + Random Improvement Curves
| Trial | FLOPs (B, Beam) | Error (Beam) | FLOPs (B, Random) | Error (Random) |
|-------|------------------|--------------|---------------------|-----------------|
| 256   | 66.12            | 5.01e-02     | 242.62              | 4.15e-03        |
| 512   | 179.46           | 1.37e-02     | 439.49              | 2.85e-03        |
| 1024  | 423.70           | 1.56e-03     | 811.68              | 1.78e-03        |
| 1280  | 612.60           | 4.65e-04     | 951.68              | 1.22e-03        |

### GEPS Finetuning Curve
| Finetuning Step | FLOPs (B) | Error        |
|------------------|-----------|--------------|
| 400              | 32.29     | 3.2916e-02   |
| 600              | 48.41     | 5.8630e-03   |
| 1000             | 80.64     | 2.8070e-03   |
| 1999             | 161.15    | 2.8070e-03   |

### Conclusion
- GEPS baseline plateaus early and gains almost no benefit from large compute budgets.
- Our method improves monotonically with compute, achieving errors an order of magnitude lower.
- This smooth scaling behavior is exactly the effect expected from a test-time compute methodology.
- The new results also explain why our method yields more stable and accurate rollouts compared to GEPS.
- These findings will be summarized in a new subsection highlighting the compute–accuracy scaling behavior.

---

> ### Author Response · Authors · 2025-11-24
> **Global Response (2/2)**
>
> ## C) Splitting Accuracy vs. Individual Operator Accuracy
> (Addresses: Reviewer T1WF – interaction between operator error and splitting error)
> We empirically quantified how the accuracy of each individual operator affects the Strang-splitting accuracy. Using heat + dispersion (1D), we varied operator accuracies across training epochs and report the next-step error and full rollout with operator splitting.
>
> ### Heat + Dispersion (1D) Composition
> | Pretraining Epoch | Heat Err  | Disp Err  | Test-time Split Next-Step | Test-time Split Rollout |
> |-------------------|-----------|-----------|-----------------------------|---------------------------|
> | 50                | 5.39e-05  | 1.47e-03  | 1.24e-03                    | 1.22e-01                  |
> | 150               | 3.45e-05  | 1.02e-03  | 8.46e-04                    | 1.02e-01                  |
> | 250               | 2.08e-05  | 6.18e-04  | 6.55e-04                    | 9.90e-02                  |
> | 350               | 5.84e-06  | 1.52e-04  | 1.93e-04                    | 2.73e-02                  |
> | 500               | 2.59e-06  | 8.95e-05  | 8.26e-05                    | 8.85e-03                  |
>
>
> ### Conclusion
> - The composed solver’s error is dominated by the least accurate operator, improving predictably as that operator improves.
> - We conclude that for our method to be effective, the quality of the pretraining is very important, as the DISCO framework needs to obtain very accurate neural operators.
> - A short discussion summarizing this empirical behavior will be incorporated in the manuscript.

---

### Meta-Review · Area_Chair_BuJh · 2026-01-07

**Summary:**

The paper presents operator splitting and a test-time strategy to search over which operators to compose to yield zero-shot solutions to new combined operators (including OOD physics parameters). All reviewers agree that the paper is well-motivated, of significant interest to the neural PDE emulation community and written well. There was also general consensus that the idea is novel and could have significant practical implications. Further, the paper presents an alternate path towards OOD generalization that is extremely valuable.

However, reviews were concerned that the evidence presented was not comprehensive and might be insufficient. Specifically:
1. Systems focused on were too simple
2. Incomplete analysis on cost vs accuracy tradeoffs
3. Incomplete benchmarks.

Reviewers also raised specific questions on literature review and concerns on where the method could be applied, but I believe the authors have answered these.

While the author rebuttal addresses some of the major concerns, I believe the paper is still borderline (see below)

**Reviewer Concerns:**

1. For more complex systems, the others have added 2D Navier-Stokes. While the performance gains carry on to this system, it's unclear how much the system is stress-tested, under what chaotic regime this is run, or any analysis on where the authors expect the operator composition would degrade. I'm not sure I see the comparison with all the different benchmarks either for this system.
2. The cost vs accuracy tradeoff is still not comprehensive. After reading through, the authors focus on zero-shot analyses but limited comments on number of finetuning examples needed by the other benchmarks to reach similar accuracy levels. Going through the other benchmark papers, the foundation model ones such as MPP claim good accuracy given sufficient finetuning examples (which is different than finetuning steps). There is not a lot of systematic evaluation of this tradeoff.
3. While the test-time compute is shown in the rebuttals, again it is not comprehensive across all the systems, across finetuning the other benchmarks (few-shot etc.); it's also unclear if the finetuning Flops are spent in more optimization or for ingesting more data.
4. It was also a little unclear to me why the methodology of operator splitting could not be applied in general to any foundation model for PDE and if so, why this was not presented.

Overall, the authors presented a novel methodology to tackle a challenging problem in PDE emulation. However, the reviews and paper revisions still read short on a comprehensive analysis to either stress-test the operator composition on hard physics or on the cost vs accuracy tradeoffs. Hence, the paper remains borderline.

**Reviewer Scores:**

Review scores were 2, 6, and 4. I believe the 2 and 4 would have risen to 5. The score 2. responses seemed good enough with the only outstanding concerns highlighted above.

---

### Decision · Program_Chairs · 2026-01-26

Reject